# Inference of transmission dynamics and retrospective forecast of invasive meningococcal disease

Jaime Cascante-Vega[1¤]*, Marta Galanti[1], Katharina Schley[2], Sen Pei[1], Jeffrey Shaman[1,3]*

**1** Department of Environmental Health Sciences, Mailman School of Public Health, Columbia University, New York, New York, United States of America, **2** Pfizer Pharma GmbH, Berlin, Germany, **3** Columbia Climate School, Columbia University, New York, New York, United States of America

¤ Current address: Center for Genomics and System Biology, Department of Biology, New York University, New York, United States of America
* jaime.cascante.vega@nyu.edu (JCV); jls106@cumc.columbia.edu (JS)

**Data Availability Statement:** All the code, and scripts to download the data and reproduce the manuscript analysis as well as the Figures is available in the Github repository https://github.

## Abstract

The pathogenic bacteria *Neisseria meningitidis*, which causes invasive meningococcal disease (IMD), predominantly colonizes humans asymptomatically; however, invasive disease occurs in a small proportion of the population. Here, we explore the seasonality of IMD and develop and validate a suite of models for simulating and forecasting disease outcomes in the United States. We combine the models into multi-model ensembles (MME) based on the past performance of the individual models, as well as a naive equally weighted aggregation, and compare the retrospective forecast performance over a six-month forecast horizon. Deployment of the complete vaccination regimen, introduced in 2011, coincided with a change in the periodicity of IMD, suggesting altered transmission dynamics. We found that a model forced with the period obtained by local power wavelet decomposition best fit and forecast observations. In addition, the MME performed the best across the entire study period. Finally, our study included US-level data until 2022, allowing study of a possible IMD rebound after relaxation of non-pharmaceutical interventions imposed in response to the COVID-19 pandemic; however, no evidence of a rebound was found. Our findings demonstrate the ability of process-based models to retrospectively forecast IMD and provide a first analysis of the seasonality of IMD before and after the complete vaccination regimen.

## Author summary

This paper presents a time series analysis of Invasive Meningococcal Disease caused by the pathogenic bacteria *Neisseria meningitidis* as well as the development and validation of a suite of models for simulating and forecasting disease outcomes in the United States. We present the models and their performance and construct a multi-model ensemble (MME) forecast that combines individual models based on past forecast performance, as well as an equally weighted aggregation. Deployment of a complete vaccination regimen, introduced for adolescents in 2011, coincided with a change in the periodicity of IMD,

com/ChaosDonkey06/IMD_
RetrospectiveForecasting.

**Funding:** This study was funded by Pfizer Inc. The funders had no role in study design, data collection and analysis, decision to publish, or preparation of the manuscript.

**Competing interests:** I have read the journal's policy and the authors of this manuscript have the following competing interests: JS and Columbia University disclose partial ownership of SK Analytics. JS discloses consulting for BNI. KS is an employee of Pfizer Pharma and may hold stocks or stock options.

suggesting altered transmission dynamics. We found that a model forced with the period obtained by local power wavelet decomposition best fits and forecasts observations. In addition, the MME forecasts performed the best across the entire study period. Finally, our study included US-level data until 2022, allowing study of a possible IMD rebound after the relaxation of non-pharmaceutical interventions imposed in response to the COVID-19 pandemic; however, no evidence of a rebound was found. Our findings demonstrate the ability of process-based models to retrospectively forecast IMD and provide a first analysis of the seasonality of IMD before and after the complete vaccination regimen.

## Introduction

Invasive meningococcal disease (IMD) is caused by the bacterium *Neisseria meningitidis* (*N. meningitidis*). IMD has a rapid progression and can cause pneumonia, meningitis, and bloodstream infection. The case-fatality rate of IMD is estimated between 10% and 15%, and 20% of individuals who survive infection have lifelong disabilities, including vision and hearing loss, neurological deficits, and limb loss [1,2]. While meningococcal disease affects all age groups, infections are reported predominantly in infants, young adults, and adults over 85 years old [3], and, in temperate regions, cases are predominant in winter and spring months [4].

Nasopharyngeal colonization with *N. meningitidis* in healthy individuals is relatively common: reports show 5% to 35% of the population are carriers [5,6]. The frequency of carriage depends on age and peaks in young adults [3]. In the vast majority of cases, carriage is harmless to the host, but in some instances, shortly after colonization, the pathogen enters the bloodstream and causes invasive disease [5]. Transmission of *N. meningitidis* to a susceptible individual happens through contact with the respiratory droplets or saliva of a colonized or infected host. Due to the sensitivity of the bacteria to atmospheric conditions, transmission requires close contact [7].

There are 12 identified serogroups of *N. meningitidis*, but only 5 of them—A, B, C, W, Y—are responsible for almost all cases of invasive disease. Two types of vaccines are currently available in the United States and recommended by the US Advisory Committee on Immunization Practices (ACIP) of the Centers for Disease Control and Prevention (CDC): Meningococcal conjugate (MenACWY) vaccine, routinely recommended for primary immunization in 11–12 year olds since 2005 [8] followed by a booster at 16 years of age, which was introduced in 2011 [9]; and vaccines that protect against serogroup B meningococcal (MenB) bacteria and are recommended as a shared-clinical decision making recommendation in 2015 [10]. Rates of meningococcal disease have declined in the US during the last two decades and have remained low in recent years (0.11 cases per 100,000 population in 2019) [11]. Given the severity of IMD, it is extremely important to monitor its epidemic trends and to identify changes in carriage prevalence and vaccination rates that might lead to rapid disease resurgence.

Here we develop a suite of mechanistic and statistical models to simulate the transmission of *N. meningitidis* and forecast IMD incidence. We evaluate the retrospective accuracy of forecasting IMD at the country scale using case reports from 2006 to 2020 for the US. Specifically, we show that a model-inference system based on a combination of mechanistic models and Bayesian inference methods successfully captures IMD dynamics during the last 14 years in the US and is able to forecast future disease outcomes. Similar model-inference systems have been used for parameter estimation, evaluation of counterfactual interventions, and forecast for a variety of diseases, including influenza, SARS-CoV-2, West Nile Virus [12], malaria, dengue [13], and methicillin-resistant Staphylococcus aureus [14–20].

The optimized models developed here can be used to forecast IMD cases at the country scale and to estimate the effects of control measures, human behavior, or pathogen biology, such as drops in vaccine uptake, changes in mixing patterns across a population, or the emergence of a more virulent meningococcal strain. The present analysis is intended as validation and assessment of the *N. meningitidis* model performance, which will be leveraged in future work to study the impact of the SARS-COV-2 pandemic and vaccinations on IMD incidence.

## Materials and methods

### Data description

Weekly IMD incidence data at the state and national level in the US were compiled from the CDC Wonder dataset [21]. We used data from 2006 to 2022, as only yearly cumulative counts were reported before 2006. Classification of cases by serogroup (ACWY and B) only began in 2020 and was not used in this study; rather, incidence includes all serogroups.

### Spectral analysis

We used wavelet time series analysis to capture the temporal properties of IMD in the US [22]. The objective of this analysis is to represent the IMD time series in both the time and frequency domains and reveal shifts in seasonality or other periodicities. We used the Morlet wavelet function as the basis. Similar analyses have been previously used to explore the seasonality of measles and influenza [23]. We investigate the fit of the inverse wavelet transform (IWT), as well as study the periodicity of the system averaging the local wavelet power spectrum (LWPS) across the study period.

### Model description

We developed three mathematical mechanistic models and one purely statistical model–the autoregressive integrated moving average model (ARIMA). The three mechanistic models aim to represent the underlying transmission process of the disease, have the same core model structure shown in Fig 1, and are described in detail in S1 Text. Briefly, for the mechanistic models, we compartmentalize the population (N) into 3 groups: susceptible/non-carrier individuals (S), carriers of the bacteria (C) that are colonized with *N. meningitidis* but not infected with IMD, and carriers who have become infected with IMD (I) (Table 1). Fig 1 shows the transition between the compartments: susceptible individuals (S) become carriers with a force of infection $\lambda$ (S→C), a fraction $\theta$ of carriers become infected with IMD at the rate $\alpha_2$ (C→I) and the proportion of C that don't develop infection become susceptible again with the decolonization rate $\alpha_1$ (C→S). We model the force of infection $\lambda$ using the law of mass action with contact rate $\beta$, and assume both carriers (C) and infected (I) contribute to transmission: $\lambda = \beta (C+I)/N$. Description of the parameters, units and value used in presented in Table 2.

We segregated this core form into three separate, mechanistic models representing alternate transmission dynamics: **i)** a constant contact rate, $\beta = \beta_0$, **ii)** a seasonally varying contact rate, $\beta = \beta(t)$, and **iii)** a seasonally varying likelihood of infection $\theta = \theta(t)$. Models **ii** and **iii** assume different mechanisms behind the observed seasonality of IMD. Model form **ii** assumes that the mechanism driving seasonality is related to the transmission rate or the pattern of contact between non-carriers and carriers. Model form **iii** assumes seasonality in the likelihood of transitioning from carriage to infection, i.e. the probability $\theta$ that a given carrier becomes infected with IMD. This assumption is supported by evidence that unlike disease prevalence, carriage prevalence does not show a seasonal pattern [3,24].

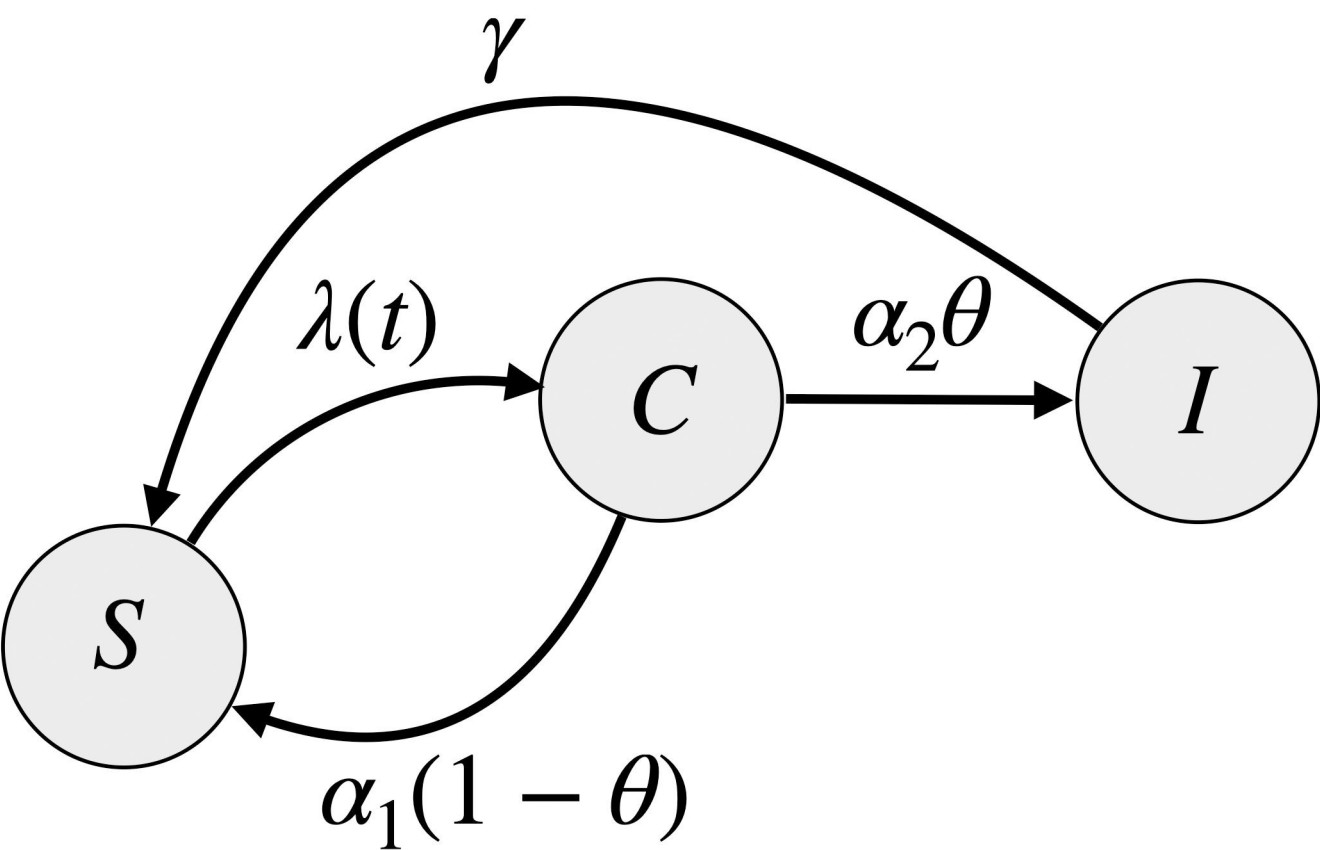

**Fig 1. Model diagram.** Compartmental model of Neisseria Meningitidis transmission. S is the susceptible/non-carrier population, C is the carrier population, and I is the infected (IMD) population. $\gamma$ is the rate of recovery after infection, $\theta$ the likelihood of infection given carriage, $\alpha_1$ is the decolonization rate and $\alpha_2$ is the infection rate. Further information on parameter ranges and units can be found in Table 2; variables are consigned in Table 1.

We computed the average power across the study period from the wavelet analysis and used the frequency with the maximum power to impose seasonality in models **ii** and **iii** (see Fig B in S1 Text lower subplot).

### Model-inference framework

The three mechanistic models (i-iii) were coupled with Bayesian inference methods that assimilate IMD surveillance data into the model simulations. For each model, an ensemble of simulations was first initialized with parameters and state variables values drawn from ranges consistent with observations and estimates reported in previous studies. Specifically, for the initialization step, we derived a prior range of parameters and variables by imposing at equilibrium that the basic reproductive number $R_0 \sim 1$ and the carriage prevalence is around 20%, we study the sensitivity of the restriction to different decolonization periods (Fig A in S1 Text). As

**Table 1. Variables of the transmission dynamical model, description and its units.**

| Variable | Description | Units |
|---|---|---|
| *S* | Susceptible individuals. | Number of Individuals |
| *C* | Carriers of *Neisseria Meningitis* | Number of Individuals |
| *I* | Infected individuals. | Number of Individuals |

**Table 2. Parameters of the transmission dynamical model, descriptions, range or value used and its units.**

| Parameters | Description | Range | Units |
|---|---|---|---|
| $\theta$ | Probability of disease given carriage. | Estimated | - |
| $\beta$ | Population-level contact rate. | Estimated | 1/days |
| $\gamma$ | Recovery rate after infection. | [7,15] | Days |
| $\alpha_1$ | Decolonization rate | [120,180] | Days |
| $\alpha_2$ | Infection rate. | [7,15] | Days |
| $\mu$ | Birth/death rate | 2.74e-5 | 1/days |

the ensembles were integrated through time, a statistical filter was used to iteratively assimilate monthly observations and adjust the prior, model-simulated distribution of variables and parameters into posterior distributions that better represent the observed dynamics. We used two alternative data assimilation (Bayesian inference) algorithms: 1) a single run of the Ensemble Adjustment Kalman Filter (EAKF) and 2) an iterated Filtering framework (IF-EAKF) [4,5], which gradually adjusts the parameters through multiple iterations of the EAKF and provide point estimates of the parameters (see S1 Text section The ensemble adjustment Kalman filter for details).

We tested each model-inference combination by 1) evaluating the posterior fit of the observable variable and 2) comparing free simulations of IMD incidence, run with the posterior parameters estimated with either EAKF or IF-EAKF, to the observed trajectory of cases (Figs C-E in S1 Text). The free simulations with the IF-EAKF point estimated parameters matched the IMD trajectory better than those generated with the EAKF alone, except for model 2 (See Figs C2-C3 and D2-D3 and E2-E3 in S1 Text for the 3 models, respectively).

## ARIMA

We optimized an autoregressive integrated moving average (ARIMA) model by computing both autocorrelation and partial autocorrelation with a maximum lag of 40 months. The model and model-fit are described in section 'The ARIMA model' in the S1 Text. We optimized the number of lag observations in the model, i.e., lag order $p$, and the size of the moving average window, i.e. order of moving average $q$, using the significant ($p < 0.05$) lags from the autocorrelation and partial autocorrelation. We fixed the number of times that the time series is differenced, $d$, as 1 (See SI section The ARIMA for details in the implementation).

## SARIMA

We optimized a seasonal autoregressive integrated moving average (SARIMA) model by computing both autocorrelation and partial autocorrelation with a maximum lag of 20 months. The model and model-fit are described in section 'The SARIMA model' in the S1 Text. We optimized the number of lag observations in the model, i.e., lag order $p$, and the size of the moving average window, i.e. order of moving average $q$, using the significant ($p < 0.05$) lags from the autocorrelation and partial autocorrelation. We fixed the number of times that the time series is differenced, $d$, as 1. We used the same order of seasonal moving average Q and seasonal lag P, as obtained from the autocorrelation and partial autocorrelation (See S1 Text section The ARIMA for details in the implementation).

## Mechanistic retrospective forecasting framework

We evaluated the forecasting skill of the different models by generating retrospective forecasts of monthly IMD cases between 2006 and 2020. Specifically, we sequentially assimilated IMD

incidence data within the EAKF framework to generate posterior fits up to each (monthly) forecast initiation date and then integrated the model into the future to generate probabilistic forecasts without further training. We also used the IF-EAKF framework to estimate parameters before initiating the forecasts and found that the EAKF alone performs better. We evaluate the forecast by plotting a subset in Fig F in S1 Text

### Evaluation of retrospective forecasting

We used one evaluation metric for point predictions—mean absolute error (MAE) and one proper scoring rule to evaluate probabilistic predictions—the Weighted Interval Score (WIS) [25]. We examined the forecast accuracy of predictions for monthly IMD cases 1 to 6 months in the future. MAE is calculated as the absolute value of the difference between the mean prediction of the probabilistic forecast and reported IMD incidence. The WIS accounts for the probabilistic distributions of predicted values specified by 20 quantile intervals [25] as described in the S1 Text (See S1 Text on the WIS computation)

### Multi-model ensemble of forecasting models and evaluation

Aggregating probabilistic forecasts generated by different model systems in a multi-model ensemble (MME) often produces more accurate 'multi-model' predictions than the individual component model systems [26,27]. We used two methods to aggregate the forecasts from different models into MME predictions. **i)** We equally weighted each model (a simple average for each quantile). **ii)** We used an expectation maximization (EM) algorithm based on a probabilistic marginal distribution to draw from the model space [26]. Here we considered two approaches for computing the marginal distribution of each model forecast for the EM algorithm: **a)** the WIS of each model for all prior (historical) predictions and **b)** the WIS of each model for a fixed window (K months) of past predictive performance. See S1 Text for further information on the implementation of the MME methods. We examined the performance of the MME predictions using the WIS, as described for the component models.

Transmission dynamics during the study period were possibly impacted by changes in vaccination policy, specifically, the introduction of a booster shot for 16-year-olds after 2011. To account for this exogenous factor, we investigated the performance of the models during three different study periods: the entire study period, before 2011, and after 2011.

## Results

### Wavelet time-series analysis

A local wavelet power spectrum (LWPS) of the weekly national IMD incidence time series for the US is shown in Fig 2. The inverse transform is presented in Fig A in S1 Text. The start of the complete vaccination regimen in adolescents (primary dose + booster) is indicated by the vertical line corresponding to year 2011 [9]. Prior to that, the LWPS shows consistently high power at 1-year and 0.3-year (4-month) periodicities. After introduction of the complete adolescent vaccination program, the period with maximum power decreases to around 0.9 years and the magnitude of the maximum power decreases. After 2011, the power decreased consistently until the end of the study period (see Fig 2C at a period equal to y = 1.1 years).

Fig B in S1 Text shows the average power across the study period, which maximizes at 0.98 years. This periodicity was used to modulate the contact rate and the likelihood of infection in mechanistic model structures **ii** and **iii**, respectively (see Methods section).

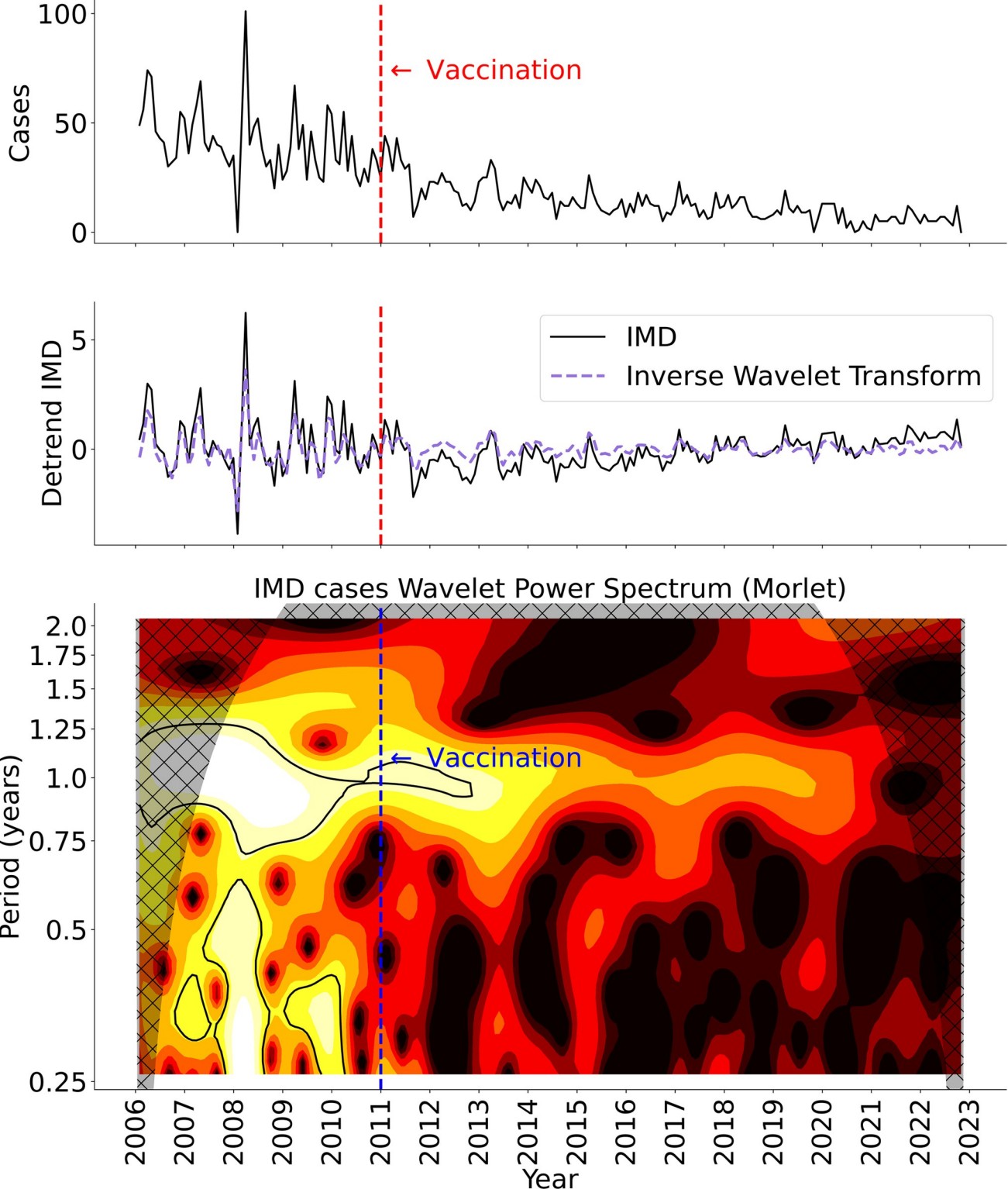

**Fig 2. Wavelet timeseries analysis. A)** Time series of Invasive Meningococcal disease (IMD) monthly incidence for the US, 2006–2022 **B)** Normalized detrended IMD incidence (black line) and inverse wavelet transform (salmon dashed line) **C)** Local wavelet power spectrum; power is color coded with lower magnitudes shown in darker red and higher magnitudes in lighter yellow.

## Posterior fit and free simulation with MLE

We evaluated the posterior fit of the model-inference framework for the three mechanistic model structures. Model **iii** (seasonality in the likelihood of infection given carriage) consistently performed better both for its EAKF posterior incidence fit and in free simulations run with estimated parameters (Fig 3C and Figs C-E in S1 Text.). The posterior incidence of model iii) simulates IMD data well across the entire study period except for the spike with exceptionally high reported levels of IMD during February 2008. The EAKF posterior estimates for all 3 models are shown in Figs E and H and K in S1 Text. For all models, the posterior susceptibility profile increases as a function of time, but the posterior estimates of model **iii** capture observed seasonal patterns. In particular, free simulation using the posterior estimates of the IF-EAKF spans observed IMD incidence (Figs C-E in S1 Text), indicating that the model can reproduce transmission dynamics.

We estimated that before the start of the complete vaccination regimen in 2011 carriage prevalence was 6.00% (95% CI: 3.20–10.29%); after 2011, this estimate dropped to 1.64% (95% CI: 1.18–2.18) (Fig 3, right plot). The prevalence of the infected population was 0.019 (95% CI: 0.014–0.025) per 100,000 population before 2011 and dropped to 0.0072 (95% CI: 0.0049–0.0098) per 100,000 population after 2011 for the US, following the same decreasing trend.

## Retrospective forecasting using individual models

We generated retrospective forecasts using the four individual models in order to quantify the performance of each model using out-of-sample predictions (Fig F in S1 Text). In general, model **iii** possesses a narrower prediction interval than the purely statistical autoregressive ARIMA. Relative performance across models remained similar as the forecast horizon increased from 1 to 6 months. Point predictions across the study period indicate that the mechanistic models consistently outperformed the ARIMA and that differences among the three mechanistic models were negligible, consistent across forecast horizons (Figs G-H in S1 Text). The mean error for each forecast date showed substantial underprediction of the unusual peak during February 2008. The overbroad probabilistic forecasts of the ARIMA and SARIMA were penalized by the WIS score, resulting in higher WIS scores than the mechanistic models (i.e., a worse performance). Finally, we compared the mean performance of the models across the entire study period and before and after the beginning of the complete adolescent vaccine regimen in 2011. Overall, the mechanistic models outperformed the ARIMA and SARIMA considering the average performance during the study period (Fig I1 in S1 Text); we used the Wilcoxon signed-rank test to assess statistical of the distribution of WIS between the ARIMA and each mechanistic model and conclude that mechanistic models performed better, and this finding was consistent across forecast horizons (see Tables A-B in S1 Text for the p-values and Fig I2 in S1 Text for the WIS distribution). Among the three mechanistic models, model **iii** outperformed the others in the average performance during the study period; however, model **i** produced better forecasts for the period prior to 2011 (Fig I in S1 Text).

## Forecasting with a multi-model ensemble (MME)

We found that MME constructed using all past predictions performed better across all periods and horizons than MME constructed using only more recent predictions. We also found that forecast performance worsened (WIS increases) as the size of the training window increased (Fig J in S1 Text). Finally, we compared the best individual model **iii**, the trained MME using performance from the preceding 2 months, the MME using all past predictions, and the equally weighted MME (see Fig 4). We found that the MME constructed with all past

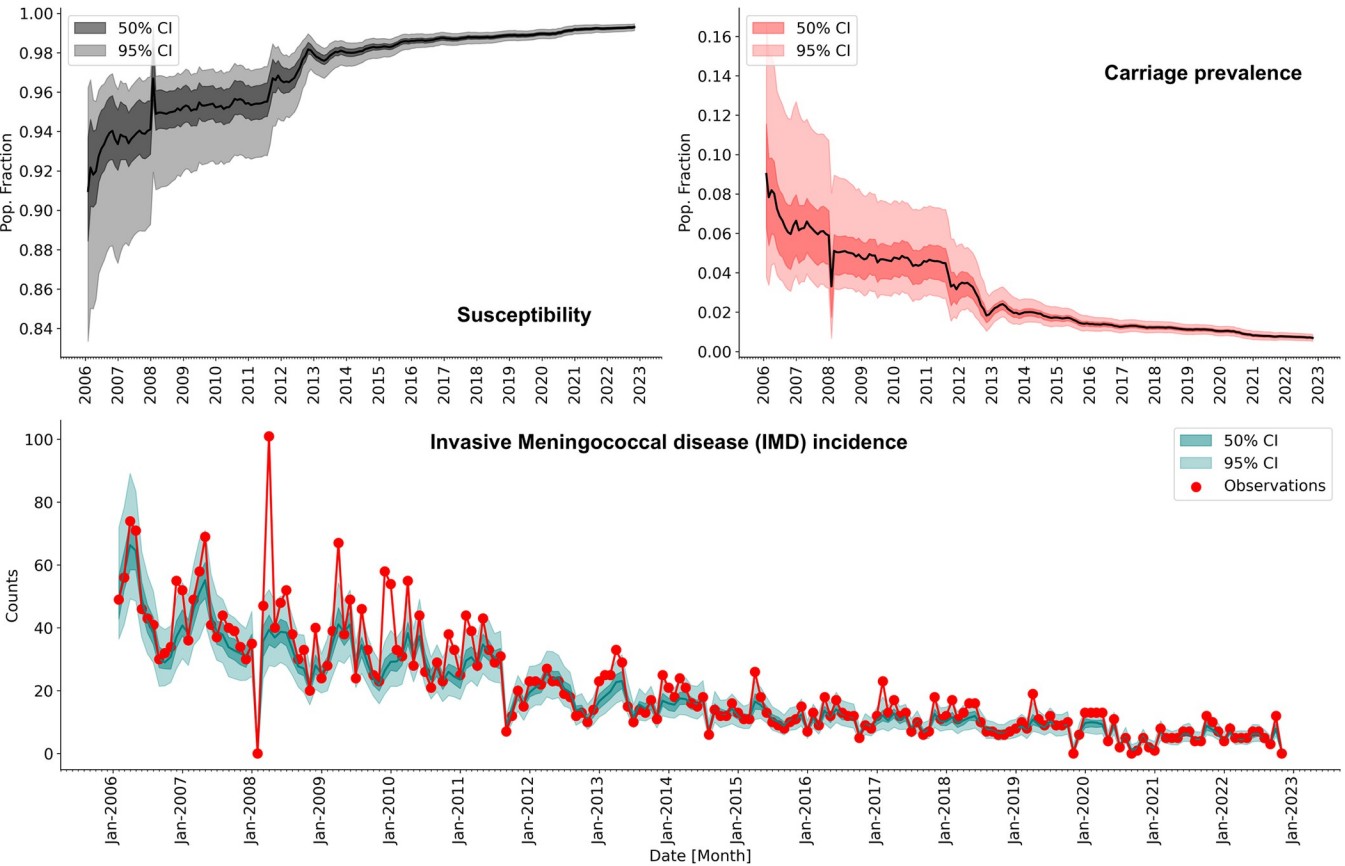

**Fig 3. Posterior fit. Model iii) (seasonality in the likelihood of infection). A)** Simulated susceptibility; the solid line shows the median and the darker and lighter ribbons show 50% and 95% CI. **B)** Carriage prevalence; the solid line shows the median and the darker and lighter ribbons show 50% and 95% CI. **C)** Invasive Meningococcal Disease (IMD) incidence. The teal line, darker and lighter ribbons show the simulated median and 50% and 95% CI, respectively. Red dots are observations.

predictions outperformed the other MME approaches for the entire study period; however, for the sub-periods before and after the start of the complete vaccine regimen, mean WIS was lower for both model **iii** and the MME trained with 2 months of prior performance. The untrained MME had the worst performance across all data splits (Fig 4). To further examine statistically significant differences, we plotted the distribution of WIS and used the Wilcoxon signed-rank test to assess differences, we only found statistical significance differences between the equally weighted ensemble and the rest of the models, model 3, MME with 2 months and all past performance (See Table C and Fig K in S1 Text). We also compared the performance of the MME after 2011 (onset of the complete vaccine regime) and found that the MME trained with performance during the prior 2 months was the best across all past performances. Finally, to understand the importance assigned to each model we present the weights from the MME (Fig M in S1 Text in general, the MME principally weighted the mechanistic models with the greatest weight alternating among model 1 and model 3 for some study periods.

## Discussion

In this work we develop and test different model structures for reproducing and forecasting invasive meningococcal disease dynamics, resulting from infection by the bacterium *Neisseria meningitidis*. We used IMD incidence in US from 2006 to 2022 to calibrate and test the

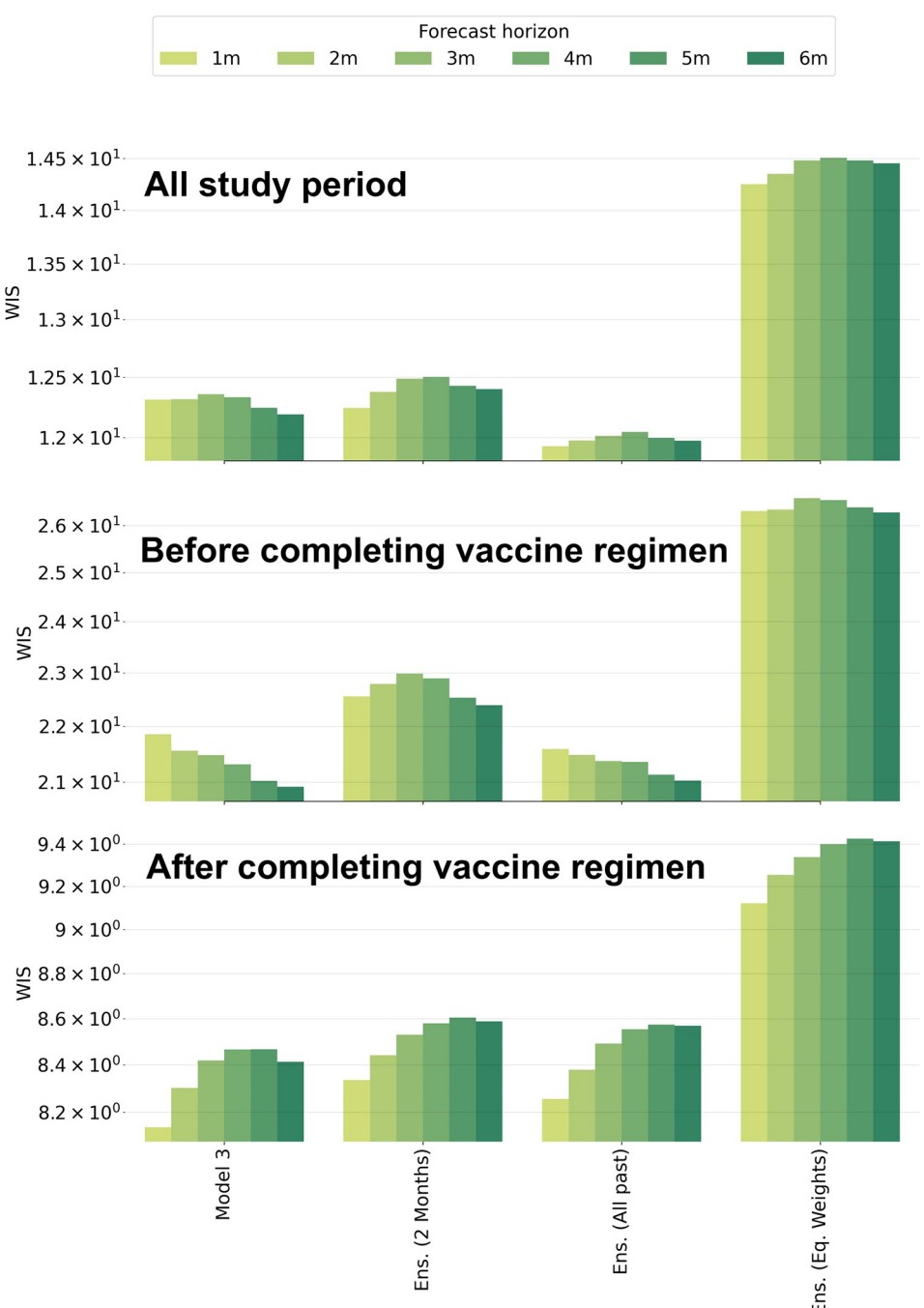

**Fig 4. Performance of model and ensemble forecasts.** Each bar shows the mean performance for the indicated split of the data; forecast horizon is color-coded and indicated in the legend. **A)** The entire study period **B)** Pre 2011. **C)** Post 2011.

forecasting skill of 5 different models and their ensemble combinations. The findings provide a foundation for conducting future analyses to investigate the impact of vaccination and changes in mixing patterns resulting from the impact of the SARS-COV-2 pandemic.

Here, we found a marked seasonality of IMD before 2011 with a 1-year period followed by a decreasing seasonal signal after 2011. This qualitative change coincided with changes in

vaccine policy and uptake: teenager vaccination for *N meningitidis* was introduced in 2005 and was extended with an additional booster in 2011. It is possible that vaccination contributed to this shift in seasonality and to the decrease in overall IMD incidence observed in the last 2 decades (11). Evidence suggests that the ACWY conjugate vaccine reduces carriage [28], impacting transmission dynamics by reducing the force of infection. However, it is also possible that the observed change in seasonality is explained by changes in the age distribution of infection [29].

We explored different mechanisms through which seasonality could affect transmission by testing different transmission model forms. Different climatic, socio-demographical, and behavioural factors may affect transmission in different ways and drive seasonal patterns. Influenza transmission is modulated by absolute humidity [14,30,31,31]; malaria transmission is modulated by rain, temperature, and humidity [32]; dengue, and other arboviruses outbreaks are modulated by the synergistic effects of temperature and population density [33,34]; human mobility shapes SARS-CoV2 transmission [35]; and rainfall modulates cholera dynamics [36–38]. Researchers can represent these drivers by designing mathematical models in which the contact or transmission rate is a function of these relevant factors. Here we show that forcing the IMD transmission model via a seasonal likelihood of infection given carriage better explains observed incidence and had the best performance in retrospective forecast (Figs K-M in S1 Text). This finding is potentially supported by carriage surveys studies in the meningitis belt of Sub-Saharan Africa, that have reported a large increase of the disease to carrier ratio during the dry season [39]. Despite the different climate conditions in the US, evidence from the meningitis Belt supports a potential effect of climate on disease mechanisms.

This result suggests important future areas for research to improve understanding of the mechanisms behind this forcing (e.g. climatic, contact, phenotypical, etc) [4,24]. That is, while IMD rates display seasonal trends, peaking in winter months, carriage prevalence does not show seasonality [4]. There are several mechanisms that might increase the likelihood of infection with IMD in winter months: 1) cold/dry air can damage the nasopharyngeal mucosa of the host facilitating bacterial invasion, as proposed by studies set in the meningitis belt of Sub-Saharan Africa, where dry Harmattan winds are believed to be responsible for increased disease-to-carriage ratios [40]; 2) previous infections with some seasonal viruses (i.e. influenza) can predispose the host to IMD infection; and 3) seasonal factors may affect the host immune system, making some more prone to disease [4]. However, current IMD data availability limits the possibility of investigating these mechanisms. Stratification of the IMD incidence dataset by age, serogroup and vaccination status could support testing these hypotheses through more detailed IMD modeling. Lastly, future research, perhaps leveraging a more complete IMD dataset resolved at finer spatial scales, could shed some light on the link between influenza and IMD. If climate is shown to be a driver of the seasonality observed [41,42], the effect of a changing climate would need to be incorporated into IMD models.

We also reviewed the literature for other possible determinants of transmission and found there have been multiple outbreaks of IMD among men who have sex with men (MSM) in the last 20 years. Transmission of IMD in the MSM community requires further study. In our mechanistic models, we assume homogeneous mixing of the population, so specific mixing patterns among subpopulations are not represented; however, the model could be elaborated in the future to represent subpopulations. Additionally, we do not explicitly model the effect of vaccination and its possible impact on the carriage acquisition or the likelihood of infection. It also has been shown that prior influenza infection is a risk for IMD [24], which could also make model **iii** the best just by seasonally adjusting the likelihood of infection $\theta$.

The posterior estimates of susceptibility and carriage for the 3 models (Fig 3C and Figs C-E in S1 Text) show that susceptibility increased during the study period, and in consequence,

carriage decreased. This result could be due to the combined effects of immunity acquired via infection and vaccination. Additionally, the posterior estimates for all models show a substantial drop in carriage during 2011 that we think is a consequence of the introduction of a booster in 2011 combined with immunity acquired via natural infection. We also investigated the effect of system initial conditions, which assumed prevalence between 5–30% (see S1 Text Equilibrium section and Fig A in S1 Text), on posterior estimates. We found that the posterior estimates remained unchanged, suggesting the system is correctly identifying susceptibility and prevalence. Our models also estimated a low fraction of the population as susceptible by the end of the data record, suggesting that rebounds caused by possible increases in susceptibility during 2020 due to non-pharmaceutical intervention to control the spread of SARS-CoV2 were not of substantial magnitude. However, our modelling approach does not account for spatial heterogeneity within the US. As a consequence, we cannot describe any geographical spots that might have a substantial pocket of susceptibles and therefore be where IMD rebound might be probable. A modeling study from the UK, accounting for the effects of decreased vaccination during the pandemic, suggests a long term effect of NPIs on carriage prevalence [43]. However, further modeling, validated with recent, local data are needed to better assess the effects of the COVID-19 pandemic on IMD.

The model-inference structures developed here can retrospectively predict the transmission of *Neisseria meningitidis* in the continental US (See Fig 4, and Figs G-I in S1 Text) We showed that in all periods the purely statistical models, ARIMA and SARIMA, performed the worst, whereas mechanistic models 1 and 3 were best across study periods (Fig I in S1 Text). In addition to individual models, we evaluated an MME forecasting system comprised of four models —one statistical and three mechanistic. For the MME forecasting system based on the past performance of the individual models, the form using all past predictions for establishing component model weights outperformed forms using only recent predictions (Figs J-K in S1 Text). This trained MME also outperforms all the individual models and the equally weighted MME (Fig 4). This result is consistent with research on endemic diseases [26]. We also showed that all individual models outperformed the equally weighted MME model (Fig 4 and Fig K in S1 Text). This finding contradicts previous research showing that equally weighted ensembles usually outperform individual models for an endemic respiratory disease [26].

Limitations in this study arise principally from the geographical resolution considered and the assumption of complete mixing across the US. Additionally, data on carriage are poor, so inference was only informed by incidence of IMD. We also didn't find any available data on vaccination with the exception of data from NIS teen surveys (Fig O in S1 Text); however, these survey data are not representative of vaccine hesitancy for the US. Limitations affecting the forecasting system include that real-time predictions are compromised by delays in reporting IMD. The implications of the change of seasonality after 2011 might be confounded by vaccination patterns or outbreaks in certain subpopulations, such as MSM [44]. The fact that influenza infection is a risk for IMD (causing a possible increase of $\theta$) is not modeled explicitly and a model representing both infections with influenza and IMD might better explain transmission dynamics [24].

## Supporting information

**S1 Text. Supplementary information with sections as follows: Description of the time-series analysis using wavelets, description of the process-based models, calculation of the disease-free equilibrium (DFE), non-DFE and basic reproductive number, description of the Bayesian inference method the Ensemble Adjustment Kalman Filter (EAKF) and a description of the retrospective forecasting and the algorithm to produce the Multi-model**

Ensemble (MME). We included 2 last sections with the Supplementary Tables and Figures, a description of these is listed below. **Table A**. Tables with Wilxonxon signed rank significant statistical tests for the ARIMA vs each individual process-based model. **Table B**. Tables with Wilxonxon signed rank significant statistical tests for the SARIMA vs each individual process-based model. **Table C**. Tables with Wilxonxon signed rank significant statistical tests for the MME vs process-based model 3. **Fig A.** Heatmap of R0 and carriage prevalence for varying values of the contact rate and likelihood of infection upon carriage. In Figs A1-A3 in s1 Text, we varied the recovery rate from 3 to 60 days, as indicated in the legend. **Fig B.** Inverse Wavelet Transform and detrended IMD incident cases. **Fig C.** C1. Posterior estimate of model 1 state variables from an EAKF. C2. Simulation of model 1 with time-varying parameters estimates from an EAKF. C3. Simulation of model 1 with point parameters estimates from an IF-EAKF. **Fig D.** D1. Posterior estimate of model 2 state variables from an EAKF. D2. Simulation of model 2 with time-varying parameters estimates from an EAKF. D3. Simulation of model 2 with point parameters estimates from an IF-EAKF. **Fig E.** E1. Posterior estimate of model 3 state variables from an EAKF. E2. Simulation of model 3 with time-varying parameters estimates from an EAKF. E3. Simulation of model 3 with point parameters estimates from an IF-EAKF. **Fig F.** Visualization of the forecast at a 6-month forecast horizon for the individual models. **Fig G.** G1. Time series of the WIS at 1-month forecast horizon for the 5 individual models. G2. Time series of the WIS at 3-month forecast horizon for the 5 individual models. G3. Time series of the WIS at 6-month forecast horizon for the 5 individual models. **Fig H.** Time series of the WIS at 6-month forecast horizon for the dynamical mechanistic models. **Fig I.** I1. Mean weighted interval score for the 5 individual models. I2. Boxplot of the WIS for the 5 individual models. **Fig J.** Mean WIS for the ensembles trained with different information of previous models' performance. **Fig K.** Boxplot of the WIS for the ensembles trained will all past performance with the previous 2 months and naively and best individual model. **Fig L.** MME weights assigned to each model trained with the performance of the previous K months. L1. K: 3 months, L2. K: 6 months, L3. K: all past performance. **Fig M.** Coverage estimates of Meningococcal vaccination for teenagers.
(PDF)

## Author Contributions

**Conceptualization:** Jaime Cascante-Vega, Marta Galanti, Sen Pei, Jeffrey Shaman.

**Data curation:** Jaime Cascante-Vega, Marta Galanti.

**Formal analysis:** Jaime Cascante-Vega, Marta Galanti.

**Funding acquisition:** Jeffrey Shaman.

**Investigation:** Jaime Cascante-Vega, Marta Galanti, Katharina Schley, Sen Pei, Jeffrey Shaman.

**Methodology:** Jaime Cascante-Vega, Marta Galanti, Sen Pei, Jeffrey Shaman.

**Project administration:** Katharina Schley, Jeffrey Shaman.

**Resources:** Jeffrey Shaman.

**Software:** Jaime Cascante-Vega, Marta Galanti.

**Supervision:** Jeffrey Shaman.

**Validation:** Jaime Cascante-Vega, Marta Galanti.

**Visualization:** Jaime Cascante-Vega.

**Writing – original draft:** Jaime Cascante-Vega, Marta Galanti.

**Writing – review & editing:** Jaime Cascante-Vega, Marta Galanti, Katharina Schley, Sen Pei, Jeffrey Shaman.

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
