## [Decision Letter · Decision Letter 0]

29 Jun 2023

Dear Mr Cascante Vega,

Thank you very much for submitting your manuscript "Inference of transmission dynamics and retrospective forecast of invasive meningococcal disease." for consideration at PLOS Computational Biology.

As with all papers reviewed by the journal, your manuscript was reviewed by members of the editorial board and by several independent reviewers. In light of the reviews (below this email), we would like to invite the resubmission of a significantly-revised version that takes into account the reviewers' comments.

We cannot make any decision about publication until we have seen the revised manuscript and your response to the reviewers' comments. Your revised manuscript is also likely to be sent to reviewers for further evaluation.

Sincerely,

Claudio José Struchiner, M.D., Sc.D.

Academic Editor

PLOS Computational Biology

Virginia Pitzer

Section Editor

PLOS Computational Biology

Reviewer's Responses to Questions

**Comments to the Authors:**

Reviewer #1: This is a well written manuscript, in which the authors set out to model invasive meningococcal disease in the US, using a simple approach based mainly on seasonality of risk of invasive disease and general carriage trends around vaccine programs.

I’m not an expert of modeling techniques, in particular not Bayesian methods, so cannot comment on the methodology used.

Overall, the authors conclude that a wavelet analysis with seasonality can reasonably well predict overall evolution of IMD case number in the US. However, the utility of such approach is less clear: how does this guide vaccine recommendation, by serogroup and by age group?

Extreme incidences like in 2008-2010 are poorly modelled. This may be explained by one major limitation of the work (recognized by the authors): the fact that the trends in influenza incidences are not included. On the same vein, it appears simplistic not to include strain evaluations or serogroup distributions in the modelling effort.

I would appreciate a discussion of predictions taking into account climate change in the US: how would this be captured by the model?

The authors argue for some assumptions with reference to evidence from the African meningitis belt. However, the role of the climate in carriage and invasive disease is probably not transposable to the situation of the US, and I would recommend being less affirmative in this perspective.

It is not clear in how far the predicted carriage prevalences are compatible with observed data – this would be important to address.

In the supplementary material, some models show an IMD peak in 2019 – is this an artefact? Please comment on this observation in the discussion.

From the manuscript, it remains unclear how the incidence of IMD changed across the Covid-19 pandemic, during and following the restrictions. This should be specifically addressed, globally in the US and regarding local outbreaks.

Finally, please provide coverage estimates for the recommended and not systematically recommended meningococcal vaccines in the US.

Reviewer #2: please see attachment

**Have the authors made all data and (if applicable) computational code underlying the findings in their manuscript fully available?**

Reviewer #1: Yes

Reviewer #2: Yes

PLOS authors have the option to publish the peer review history of their article (what does this mean?). If published, this will include your full peer review and any attached files.

Reviewer #1: No

Reviewer #2: No
---

## [Editor Report · Decision Letter 1]

2 Oct 2023

Dear Mr Cascante Vega,

We are pleased to inform you that your manuscript 'Inference of transmission dynamics and retrospective forecast of invasive meningococcal disease.' has been provisionally accepted for publication in PLOS Computational Biology.

Best regards,

Claudio José Struchiner, M.D., Sc.D.

Academic Editor

PLOS Computational Biology

Virginia Pitzer

Section Editor

PLOS Computational Biology

---

## [Editor Report · Acceptance letter]

18 Oct 2023

PCOMPBIOL-D-23-00626R1 

Inference of transmission dynamics and retrospective forecast of invasive meningococcal disease.

Dear Dr Cascante Vega,

I am pleased to inform you that your manuscript has been formally accepted for publication in PLOS Computational Biology. Your manuscript is now with our production department and you will be notified of the publication date in due course.

With kind regards,

Judit Kozma
